# PAIRWISE AUGMENTED GANS WITH ADVERSARIAL RECONSTRUCTION LOSS

## ABSTRACT

We propose a novel autoencoding model called Pairwise Augmented GANs. We train a generator and an encoder jointly and in an adversarial manner. The generator network learns to sample realistic objects. In turn, the encoder network at the same time is trained to map the true data distribution to the prior in latent space. To ensure good reconstructions, we introduce an *augmented* adversarial reconstruction loss. Here we train a discriminator to distinguish two types of pairs: an object with its augmentation and the one with its reconstruction. We show that such adversarial loss compares objects based on the content rather than on the exact match. We experimentally demonstrate that our model generates samples and reconstructions of quality competitive with state-of-the-art on datasets MNIST, CIFAR10, CelebA and achieves good quantitative results on CIFAR10.

## 1  INTRODUCTION

Deep generative models are a powerful tool to sample complex high dimensional objects from a low dimensional manifold. The dominant approaches for learning such generative models are variational autoencoders (VAEs) (Kingma & Welling, 2014; Rezende et al., 2014) and generative adversarial networks (GANs) (Goodfellow et al., 2014). VAEs allow not only to generate samples from the data distribution, but also to encode the objects into the latent space. However, VAE-like models require a careful likelihood choice. Misspecifying one may lead to undesirable effects in samples and reconstructions (e.g., blurry images). On the contrary, GANs do not rely on an explicit likelihood and utilize more complex loss function provided by a discriminator. As a result, they produce higher quality images. However, the original formulation of GANs (Goodfellow et al., 2014) lacks an important encoding property that allows many practical applications. For example, it is used in semi-supervised learning (Kingma et al., 2014), in a manipulation of object properties using low dimensional manifold (Creswell et al., 2017) and in an optimization utilizing the known structure of embeddings (Gómez-Bombarelli et al., 2018).

VAE-GAN hybrids are of great interest due to their potential ability to learn latent representations like VAEs, while generating high-quality objects like GANs. In such generative models with a bidirectional mapping between the data space and the latent space one of the desired properties is to have good reconstructions ($x \approx G(E(x))$). In many hybrid approaches (Rosca et al., 2017; Ulyanov et al., 2018; Zhu et al., 2017; Brock et al., 2017; Tolstikhin et al., 2017) as well as in VAE-like methods it is achieved by minimizing $L_1$ or $L_2$ pixel-wise norm between $x$ and $G(E(x))$. However, the main drawback of using these standard reconstruction losses is that they enforce the generative model to recover too many unnecessary details of the source object $x$. For example, to reconstruct a bird picture we do not need an exact position of the bird on an image, but the pixel-wise loss penalizes a lot for shifted reconstructions. Recently, Li et al. (2017) improved ALI model (Dumoulin et al., 2017; Donahue et al., 2017) by introducing a reconstruction loss in the form of a discriminator which classifies pairs $(x, x)$ and $(x, G(E(x)))$. However, in such approach, the discriminator tends to detect the fake pair $(x, G(E(x)))$ just by checking the identity of $x$ and $G(E(x))$ which leads to vanishing gradients.

In this paper, we propose a novel autoencoding model which matches the distributions in the data space and in the latent space independently as in Zhu et al. (2017). To ensure good reconstructions, we introduce an *augmented* adversarial reconstruction loss as a discriminator which classifies pairs $(x, a(x))$ and $(x, G(E(x)))$ where $a(\cdot)$ is a stochastic augmentation function. This enforces the

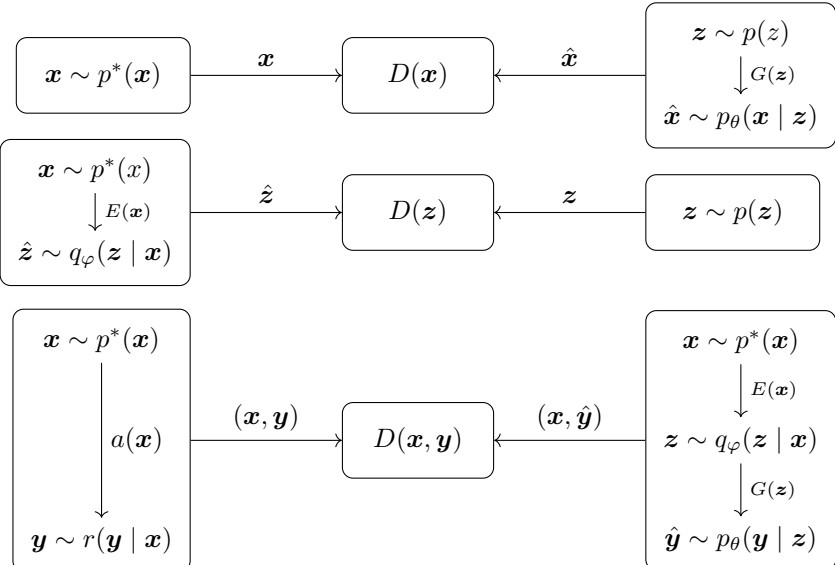

Figure 1: The PAGAN model.

discriminator to take into account content invariant to the augmentation, thus making training more robust. We call this approach Pairwise Augmented Generative Adversarial Networks (PAGANs). Measuring a reconstruction quality of autoencoding models is challenging. A standard reconstruction metric RMSE does not perform the content-based comparison. To deal with this problem we propose a novel metric *Reconstruction Inception Dissimilarity* (RID) which is robust to content-preserving transformations (e.g., small shifts of an image). We show qualitative results on common datasets such as MNIST (LeCun & Cortes, 2010), CIFAR10 (Krizhevsky et al., 2009) and CelebA (Liu et al., 2015). PAGANs outperform existing VAE-GAN hybrids in Inception Score (Salimans et al., 2016) and Fréchet Inception Distance (Heusel et al., 2017) except for the recently announced method PD-WGAN (Gemici et al., 2018) on CIFAR10 dataset.

## 2 PRELIMINARIES

Let us consider an adversarial learning framework where our goal is to match the true distribution $p^*(x)$ to the model distribution $p_\theta(x)$. As it was proposed in the original paper Goodfellow et al. (2014), the model distribution $p_\theta(x)$ is induced by the generator $G_\theta : z \to x$ where $z$ is sampled from a prior $p(z)$. To match the distributions $p^*(x)$ and $p_\theta(x)$ in an adversarial manner, we introduce a discriminator $D_\psi : x \to [0, 1]$. It takes an object $x$ and predicts the probability that this object is sampled from the true distribution $p^*(x)$. The training procedure of GANs (Goodfellow et al., 2014) is based on the minimax game of two players: the generator $G_\theta$ and the discriminator $D_\psi$. This game is defined as follows

$$\min_\theta \max_\psi V(\theta, \psi) = \mathbb{E}_{p^*(x)} \log D_\psi(x) + \mathbb{E}_{p_z(z)} \log(1 - D_\psi(G_\theta(z))) \tag{1}$$

where $V(\theta, \psi)$ is a value function for this game.

The optimal discriminator $D_{\psi^*}$ given fixed generator $G_\theta$ is

$$D_{\psi^*}(x) = \frac{p^*(x)}{p^*(x) + p_\theta(x)} \tag{2}$$

and then the value function for the generator $V(\theta, \psi^*)$ given the optimal discriminator $D_{\psi^*}$ is equivalent to the Jensen-Shanon divergence between the model distribution $p_\theta(x)$ and the true distribution $p^*(x)$, i.e.

$$\theta^* = \arg\min_\theta V(\theta, \psi^*) = \arg\min_\theta JSD(p^*(x)\|p_\theta(x)). \tag{3}$$

However, in practice, the gradient of the value function $V(\theta, \psi)$ with respect to the generator's parameters $\theta$ vanishes to zero. Therefore, Goodfellow et al. (2014) proposed to train the generator $G_\theta$ by minimizing $-\log D_\psi(G_\theta(z))$ instead of $\log(1 - D_\psi(G_\theta(z)))$. This loss for the generator provides much more stable gradients and has the same fixed point as the minimax game of $D_\psi$ and $G_\theta$.

## 3 PAIRWISE AUGMENTED GENERATIVE ADVERSARIAL NETWORKS

In PAGANs model our aim is not only to learn how to generate real objects with the generator $G_\theta(z)$ where $z$ is sampled from prior $p(z)$ but at the same time learn an inverse mapping (encoder) $E_\varphi : x \to z$. Additionally, we use the third stochastic transformation $a : x \to y$ without parameters which is called augmenter. It produces the augmentation $y$ of the source object $x$.

Let us consider the distributions which are induced by these three mappings

- $p_\theta(x|z)$ - the conditional distribution of outputs of the generator $G_\theta(z)$ given $z$;
- $q_\varphi(z|x)$ - the conditional distribution of outputs of the encoder $E_\varphi(x)$ given $x$;
- $r(y|x)$ - the conditional distribution over the augmentations $a(x)$ given a source object $x$.

Within the PAGANs model our goal is to find such optimal parameters $\theta^*$ and $\varphi*$ that ensure

1. *generator matching*: $p_{\theta^*}(x) = p^*(x)$ where $p_{\theta^*}(x) = \int p_{\theta^*}(x|z)p(z)dz$, i.e. the generator $G_{\theta^*}$ samples objects from the true distribution $p^*(x)$;
2. *encoder matching*: $q_{\varphi^*}(z) = p(z)$ where $q_{\varphi^*}(z) = \int q_{\varphi^*}(z|x)p^*(x)dx$, i.e. the encoder $E_\varphi$ generates embeddings $z$ as the prior $p(z)$;
3. *reconstruction matching*: $p_{\theta^*, \varphi^*}(y|x) = r(y|x)$ where

$$p_{\theta^*, \varphi^*}(y|x) = \int p_{\theta^*}(y|z)q_{\varphi^*}(z|x)dz, \tag{4}$$

i.e. reconstructions $G_{\theta^*}(E_{\varphi^*}(x))$ are distributed as augmentations $r(y|x)$ of the source object $x$.

### 3.1 GENERATOR & ENCODER MATCHING

In order to deal with generator and encoder matching problems we can use the framework of the vanilla GANs (Goodfellow et al., 2014). We introduce two discriminators $D_{\psi_x}$ and $D_{\psi_z}$ for two minimax games:

- generator matching:

$$\min_\theta \max_{\psi_x} V_x(\theta, \psi_x) = \mathbb{E}_{p^*(x)} \log D_{\psi_x}(x) + \mathbb{E}_{p_\theta(x)} \log(1 - D_{\psi_x}(x)) \tag{5}$$

- encoder matching:

$$\min_\varphi \max_{\psi_z} V_z(\varphi, \psi_z) = \mathbb{E}_{p(z)} \log D_{\psi_z}(z) + \mathbb{E}_{q_\varphi(z)} \log(1 - D_{\psi_z}(z)) \tag{6}$$

Then the value functions $V_x$ and $V_z$ given the optimal discriminators $D_{\psi_x^*}$ and $D_{\psi_z^*}$ are equivalent to Jensen-Shanon divergence:

$$\theta^* = \arg\min_\theta V_x(\theta, \psi_x^*) = \arg\min_\theta JSD(p^*(x) \| p_\theta(x)) \tag{7}$$

$$\varphi^* = \arg\min_\varphi V_z(\varphi, \psi_z^*) = \arg\min_\varphi JSD(p(z) \| q_\varphi(z)) \tag{8}$$

### 3.2 RECONSTRUCTION MATCHING: AUGMENTED ADVERSARIAL RECONSTRUCTION LOSS

The solution of the reconstruction matching problem ensures that reconstructions $G_\theta(E_\varphi(x))$ correspond to the source object $x$ up to defined random augmentations $a(x)$. In PAGANs model we introduce the minimax game for training the adversarial distance between the reconstructions and augmentations of the source object $x$. We consider the discriminator $D_\psi$ which takes a pair $(x, y)$ and classifies it into one of the following classes:

- the *real* class: pairs $(x, y)$ from the distribution $p^*(x)r(y|x)$, i.e. the object $x$ is taken from the true distribution $p^*(x)$ and the second $y$ is obtained from the $x$ by the random augmentation $a(x)$;

- the *fake* class: pairs $(x, y)$ from the distribution

$$p^*(x)p_{\theta,\varphi}(y|x) = p^*(x) \int p_\theta(y|z)q_\varphi(z|x)dz, \tag{9}$$

i.e. $x$ is sampled from $p^*(x)$ then $z$ is generated from the conditional distribution $q_\varphi(z|x)$ by the encoder $E_\varphi(x)$ and $y$ is produced by the generator $G_\varphi(z)$ from the conditional model distribution $p_\theta(y|z)$.

Then the minimax problem is

$$\min_{\theta,\varphi} \max_\psi V(\theta, \varphi, \psi) \tag{10}$$

where

$$V(\theta, \varphi, \psi) = \mathbb{E}_{p^*(x)r(y|x)} \log D_\psi(x, y) + \mathbb{E}_{p^*(x)p_{\theta,\varphi}(y|x)} \log(1 - D_\psi(x, y)) \tag{11}$$

Let us prove that such minimax game will match the distributions $r(y|x)$ and $p_{\theta,\varphi}(y|x)$. At first, we find the optimal discriminator:

**Proposition 1.** *Given a fixed generator $G_\theta$ and a fixed encoder $E_\varphi$, the optimal discriminator $D_{\psi^*}$ is*

$$D_{\psi^*}(x, y) = \frac{r(y|x)}{r(y|x) + p_{\theta,\varphi}(y|x)} \tag{12}$$

*Proof.* Given in Appendix A.1. ☐

Then we can prove that given an optimal discriminator the value function $V(\theta, \varphi, \psi)$ is equivalent to the expected Jensen-Shanon divergence between the distributions $r(y|x)$ and $p_{\theta,\varphi}(y|x)$.

**Proposition 2.** *The minimization of the value function $V$ under an optimal discriminator $D_{\psi^*}$ is equivalent to the minimization of the expected Jensen-Shanon divergence between $r(y|x)$ and $p_{\theta,\varphi}(y|x)$, i.e.*

$$\theta^*, \varphi^* = \arg\min_{\theta,\varphi} V(\theta, \varphi, \psi_*) = \arg\min_{\theta,\varphi} \mathbb{E}_{p^*(x)} JSD(r(y|x) \| p_{\theta,\varphi}(y|x)) \tag{13}$$

*Proof.* Given in Appendix A.2. ☐

If $r(y|x) = \delta_x(y)$ then the optimal discriminator $D_{\psi^*}(x, y)$ will learn an indicator $I\{x = y\}$ as was proved in Li et al. (2017). As a consequence, the objectives of the generator and the encoder are very unstable and have vanishing gradients in practice. On the contrary, if the distribution $r(y|x)$ is non-degenerate as in our model then the value function $V(\theta, \varphi, \psi)$ will be well-behaved and much more stable which we observed in practice.

## 3.3 TRAINING OBJECTIVES

We obtain that for the generator and the encoder we should optimize the sum of two value functions:

- the generator's objective:

$$\arg\min_\theta [V_x(\theta, \psi_x) + V(\theta, \varphi, \psi)] = \tag{14}$$

$$= \arg\min_\theta \left[ \mathbb{E}_{p_\theta(x)} \log(1 - D_{\psi_x}(x)) + \mathbb{E}_{p^*(x)p_{\theta,\varphi}(y|x)} \log(1 - D_\psi(x, y)) \right] \tag{15}$$

- the encoder's objective:

$$\arg\min_\varphi [V_z(\varphi, \psi_z) + V(\theta, \varphi, \psi)] = \tag{16}$$

$$= \arg\min_\varphi \left[ \mathbb{E}_{q_\varphi(z)} \log(1 - D_{\psi_z}(z)) + \mathbb{E}_{p^*(x)p_{\theta,\varphi}(y|x)} \log(1 - D_\psi(x, y)) \right] \tag{17}$$

---

**Algorithm 1** The PAGAN training algorithm.

$\theta, \varphi, \psi_x, \psi_z, \psi_{xx} \leftarrow$ initialize network parameters
**repeat**
    $\boldsymbol{x}^{(1)}, \dots, \boldsymbol{x}^{(N)} \sim p^*(\boldsymbol{x})$                                 $\triangleright$ Draw $N$ samples from the dataset and the prior
    $\boldsymbol{z}^{(1)}, \dots, \boldsymbol{z}^{(N)} \sim p(\boldsymbol{z})$
    $\hat{\boldsymbol{z}}^{(i)} \sim q_\varphi(\boldsymbol{z} \mid \boldsymbol{x} = \boldsymbol{x}^{(i)}), \quad i = 1, \dots, N$                   $\triangleright$ Sample from the conditionals
    $\boldsymbol{x}_{pr}^{(j)} \sim p_\theta(\boldsymbol{x} \mid \boldsymbol{z} = \boldsymbol{z}^{(j)}), \quad j = 1, \dots, N$
    $\boldsymbol{x}_{rec}^{(i)} \sim p_\theta(\boldsymbol{x} \mid \boldsymbol{z} = \hat{\boldsymbol{z}}^{(i)}), \quad j = 1, \dots, N$
    $\boldsymbol{x}_{aug}^{(i)} \sim r(\boldsymbol{y} \mid \boldsymbol{x} = \boldsymbol{x}^{(i)}), \quad j = 1, \dots, N$
    $\mathcal{L}_d^x \leftarrow -\frac{1}{N} \sum_{i=1}^N \log D(\boldsymbol{x}^{(i)}) - \frac{1}{N} \sum_{j=1}^N log\left(1 - D(\boldsymbol{x}_{pr}^{(j)})\right)$  $\triangleright$ Compute discriminator loss
    $\mathcal{L}_d^z \leftarrow -\frac{1}{N} \sum_{i=1}^N \log D(\boldsymbol{z}^{(i)}) - \frac{1}{N} \sum_{j=1}^N log\left(1 - D(\hat{\boldsymbol{z}}^{(j)})\right)$
    $\mathcal{L}_d^{xx} \leftarrow -\frac{1}{N} \sum_{i=1}^N \log D(\boldsymbol{x}^{(i)}, \boldsymbol{x}_{aug}^{(i)}) - \frac{1}{N} \sum_{j=1}^N log\left(1 - D(\boldsymbol{x}^{(j)}, \boldsymbol{x}_{rec}^{(j)})\right)$
    $\mathcal{L}_g \leftarrow -\frac{1}{N} \sum_{i=1}^N \log D(\boldsymbol{x}_{pr}^{(i)}) - \frac{1}{N} \sum_{j=1}^N \log D(\boldsymbol{x}^{(j)}, \boldsymbol{x}_{rec}^{(j)})$       $\triangleright$ Compute generator loss
    $\mathcal{L}_e \leftarrow -\frac{1}{N} \sum_{i=1}^N \log D(\hat{\boldsymbol{z}}^{(i)}) - \frac{1}{N} \sum_{j=1}^N \log D(\boldsymbol{x}^{(j)}, \boldsymbol{x}_{rec}^{(j)})$        $\triangleright$ Compute encoder loss
    $\psi_x \leftarrow \psi_x - \nabla_{\psi_x} \mathcal{L}_d^x, \ \psi_z \leftarrow \psi_z - \nabla_{\psi_z} \mathcal{L}_d^z$     $\triangleright$ Gradient update on discriminator networks
    $\psi_{xx} \leftarrow \psi_{xx} - \nabla_{\psi_{xx}} \mathcal{L}_d^{xx}$
    $\theta \leftarrow \theta - \nabla_\theta \mathcal{L}_g, \ \varphi \leftarrow \varphi - \nabla_\varphi \mathcal{L}_e$         $\triangleright$ Gradient update on generator-encoder networks
**until** convergence

---

In practice in order to speed up the training we follow Goodfellow et al. (2014) and use more stable objectives replacing $\log(1 - D(\cdot))$ with $-\log(D(\cdot))$. See Figure 1 for the description of our model and Algorithm 1 for an algorithmic illustration of the training procedure.

We can straightforwardly extend the definition of PAGANs model to $f$-PAGANs which minimize the $f$-divergence and to WPAGANs which optimize the Wasserstein-1 distance. More detailed analysis of these models is placed in Appendix C.

## 4 RELATED WORK

Recent papers on VAE-GAN hybrids explore different ways to build a generative model with an encoder part. One direction is to apply adversarial training in the VAE framework to match the variational posterior distribution $q(z|x)$ and the prior distribution $p(z)$ (Mescheder et al., 2017) or to match the marginal $q(z)$ and $p(z)$ (Makhzani et al., 2016; Tolstikhin et al., 2017). Another way within the VAE model is to introduce the discriminator as a part of a data likelihood (Larsen et al., 2015; Brock et al., 2017). Within the GANs framework, a common technique is to regularize the model with the reconstruction loss term (Che et al., 2017; Rosca et al., 2017; Ulyanov et al., 2018).

Another principal approach is to train the generator and the encoder (Donahue et al., 2017; Dumoulin et al., 2017; Li et al., 2017) simultaneously in a fully adversarial way. These methods match the joint distributions $p^*(x)q(z|x)$ and $p_\theta(x|z)p(z)$ by training the discriminator which classifies the pairs $(x, z)$. ALICE model (Li et al., 2017) introduces an additional entropy loss for dealing with the non-identifiability issues in ALI model. Li et al. (2017) approximated the entropy loss with the cycle-consistency term which is equivalent to the adversarial reconstruction loss. The model of Pu et al. (2017a) puts ALI to the VAE framework where the same joint distributions are matched in an adversarial manner. As an alternative, Ulyanov et al. (2018) train generator and encoder by optimizing the minimax game without the discriminator. Optimal transport approach is also explored, Gemici et al. (2018) introduce an algorithm based on primal and dual formulations of an optimal transport problem.

In PAGANs model the marginal distributions in the data space $p^*(x)$ and $p_\theta(x)$ and in the latent space $p(z)$ and $q(z)$ are matched independently as in Zhu et al. (2017). Additionally, the augmented adversarial reconstruction loss is minimized by fooling the discriminator which classifies the pairs $(x, a(x))$ and $(x, G(E(x)))$.

## 5 EXPERIMENTS

In this section, we validate our model experimentally. At first, we compare PAGAN with other similar methods that allow performing both inference and generation using Inception Score and Fréchet Inception Distance. Secondly, to measure reconstruction quality, we introduce *Reconstruction Inception Dissimilarity* (RID) and prove its usability. In the last two experiments we show the importance of the adversarial loss and augmentations.

For the architecture choice we used deterministic DCGAN[1] generator and discriminator networks provided by `pfnet-research`[2], the encoder network has the same architecture as the discriminator except for the output dimension. The encoder's output is a factorized normal distribution. Thus $p_\theta(x|z) = \delta_{G_\theta(z)}(x)$, $q_\varphi(z|x) = \mathcal{N}(\mu_\varphi(x), \sigma_\varphi^2(x)I)$ where $\mu_\varphi, \sigma_\varphi$ are outputs of the encoder network. The discriminator $D(z)$ architecture is chosen to be a 2 layer MLP with 512, 256 hidden units. We also used the same default hyperparameters as provided in the repository and applied a spectral normalization following Miyato et al. (2018). For the augmentation $a(x)$ defined in Section 3 we used a combination of reflecting 10% pad and the random crop to the same image size. The prior distribution $p(z)$ is chosen to be a standard distribution $\mathcal{N}(0, I)$. To evaluate Inception Score and Fréchet Inception Distance we used the official implementation provided in `tensorflow 1.10.1` (Abadi et al., 2015).

To optimize objectives (16), (14), we need to have a discriminator working on pairs $(x, y)$. This can be done using special network architectures like siam networks (Bromley et al., 1993) or via an image concatenation. The latter approach can be implemented in two concurrent ways: concatenating channel or widthwise. Empirically we found that the siam architecture does not lead to significant improvement and concatenating width wise to be the most stable. We use this configuration in all the experiments.

**Sampling Quality**
To see whether our method provides good quality samples from the prior, we compared our model to related works that allow an inverse mapping. We performed our evaluations on CIFAR10 dataset since quantitative metrics are available there. Considering Fréchet Inception Distance (FID), our model outperforms all other methods. Inception Score shows that PAGANs significantly better than others except for recently announced PD-WGAN. Quantitative results are given in Table 1. For S-VAE we report IS that is reproduced using officially provided code and hyperparameters[3]. Plots with samples and reconstructions for CIFAR10 dataset are provided in Figure 2. Additional visual results for more datasets can be found in Appendix E.3.

Table 1: Inception Score and Fréchet Inception Distance for different methods. IS and FID for other methods were taken from literature (if possible). For AGE we got FID using a pretrained model.

| Model | FID | Inception Score |
|---|---|---|
| WAE-GAN (Tolstikhin et al., 2017) | 87.7 | $4.18 \pm 0.04$ |
| ALI (Dumoulin et al., 2017) | | $5.34 \pm 0.04$ |
| AGE (Ulyanov et al., 2018) | 39.51 | $5.9 \pm 0.04$ |
| ALICE (Li et al., 2017) | | $6.02 \pm 0.03$ |
| S-VAE (Chen et al., 2018) | | 6.055 |
| $\alpha$-GANs (Rosca et al., 2017) | | 6.2 |
| AS-VAE (Pu et al., 2017b) | | 6.3 |
| PD-WGAN, $\lambda_{mix} = 0$ (Gemici et al., 2018) | 33.0 | $\mathbf{6.70 \pm 0.09}$ |
| PAGAN (ours) | **32.84** | $6.56 \pm 0.06$ |

**Reconstruction Inception Dissimilarity**
The traditional approach to estimate the reconstruction quality is to compute RMSE distance from source images to reconstructed ones. However, this metric suffers from focusing on exact reconstruction and is not content aware. RMSE penalizes content-preserving transformations while allows

---

[1]DCGAN architecture is a common choice for GANs, other works use similar architecture

[2]https://github.com/pfnet-research/chainer-gan-lib

[3]https://github.com/LiqunChen0606/Symmetric-VAE

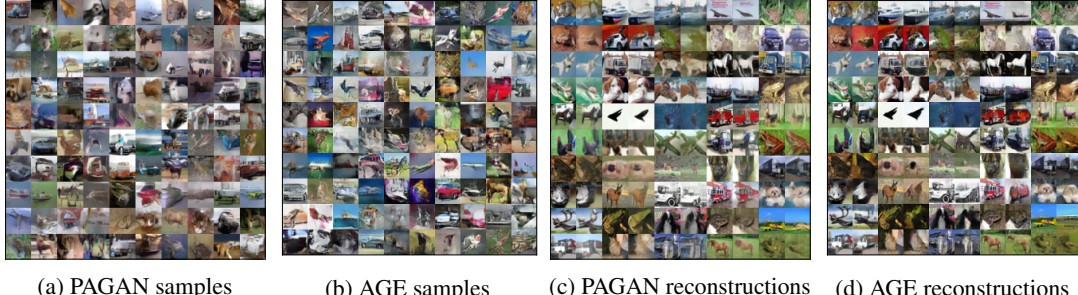

(a) PAGAN samples     (b) AGE samples     (c) PAGAN reconstructions     (d) AGE reconstructions

Figure 2: Evaluation of Generator and Encoder on CIFAR10 dataset, on plots (c), (d) odd columns denote original images, even stand for corresponding reconstructions on test partition.

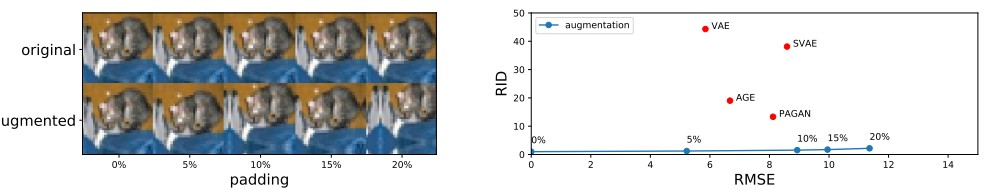

Figure 3: Reconstruction Inception Dissimilarity compared to RMSE. Unlike RMSE, RID captures distortions in image content much more better. Having same RMSE, augmentation has much more lower RID compared to a set of other methods.
* Evaluation for SVAE is based on officially provided code and hyperparameters.

such undesirable effect as blurriness which degrades visual quality significantly. We propose a novel metric *Reconstruction Inception Dissimilarity* (RID) which is based on a pre-trained classification network and is defined as follows:

$$RID = \exp\left\{\mathbb{E}_{x\sim\mathcal{D}} D_{\mathrm{KL}}(p(y|x)\|p(y|G(E(x))))\right\}, \tag{18}$$

where $p(y|x)$ is a pre-trained classifier that estimates the label distribution given an image. Similar to Salimans et al. (2016) we use a pre-trained Inception Network (Szegedy et al., 2016) to calculate softmax outputs.

Low RID indicates that the content did not change after reconstruction. To calculate standard deviations, we use the same approach as for IS and split test set on 10 equal parts[4]. Moreover RID is robust to augmentations that do not change the visual content and in this sense is much better than RMSE. To compare new metric with RMSE, we train a vanilla VAE with resnet-like architecture on CIFAR10. We compute RID for its reconstructions and real images with the augmentation (mirror 10% pad + random crop). In Table 2 we show that RMSE for VAE is better in comparison to augmented images (AUG), but we are not satisfied with its reconstructions (see Figure 12 in Ap-

Table 2: Evaluation of RMSE an RID metrics on CIFAR10 dataset.

| Model | RMSE | RID |
|-------|------|-----|
| AUG | 8.89 | $1.57 \pm 0.02$ |
| VAE | 5.85 | $44.33 \pm 2.27$ |
| SVAE | 8.59 | $38.13 \pm 1.92$ |
| AGE | 6.675 | $19.02 \pm 0.84$ |
| PAGANs | 8.12 | $\mathbf{13.01 \pm 0.82}$ |

pendix E.4), Figure 3 provides even more convincing results. RID allows a fair comparison, for VAE it is dramatically higher (44.33) than for AUG (1.57). Value 1.57 for AUG says that KL divergence is close to zero and thus content is almost not changed. We also provide estimated RID and RMSE for AGE that was publicly available[5]. From Table 2 we see that PAGANs outperform AGE which reflects that our model has better reconstruction quality.

**Importance of adversarial loss**
To prove the importance of an adversarial loss, we experiment replacing adversarial loss with the

---

[4]Split is done sequentially without shuffling
[5]Pretrained AGE: https://github.com/DmitryUlyanov/AGE

Table 3: Reconstruction Inception Dissimilarity, Inception Score and Fréchet Inception Distance calculated for three setups: 1) the proposed model, PAGAN; 2) PAGAN with $L_1$ for reconstruction loss; 3) PAGAN with augmentation removed. Model without adversarial loss or without augmentation performed worse in both generation and reconstruction tasks.

| Model | FID | IS | RID |
|---|---|---|---|
| PAGAN | 32.84 | $6.56 \pm 0.06$ | $13.01 \pm 0.82$ |
| PAGAN-L1 | 76.73 | $4.46 \pm 0.03$ | $30.94 \pm 1.58$ |
| PAGAN-NOAUG | 111.151 | $4.23 \pm 0.06$ | $50.15 \pm 2.71$ |

standard $L_1$ pixel-wise distance between source images and corresponding reconstructions and compared FID, IS and RID metrics. Using an augmentation in this setting is ambiguous. Thus we did not use any augmentation in training of the changed model. Quantitative results for the experiment are provided in Table 3. IS and FID results suggest that our model without adversarial loss performed worse in generation. Reconstruction quality significantly dropped considering RID. Visual results in Appendix E.1 confirm our quantitative findings.

**Importance of augmentation**
In ALICE model (Li et al., 2017) an adversarial reconstruction loss was implemented without an augmentation. As we discussed in Section 1 its absence leads to undesirable effects. Here we run an experiment to show that our model without augmentation performs worse. Quantitative results provided in Table 3 illustrate that our model without an augmentation fails to recover both good reconstruction and generation properties. Visual comparisons can be found in Appendix E.2. Using the results obtained from the last two experiments we conclude that adversarial reconstruction loss works significantly better with augmentation.

**Choice of Augmentation**
Experiments checking augmentation effects (see Appendix B for details) conclude the following. A good augmentation: 1) is required to be non-deterministic, 2) should preserve the content of source image, 3) should be hard to use pixel-wise comparison for discriminator.

## 6 CONCLUSIONS

In this paper, we proposed a novel framework with an augmented adversarial reconstruction loss. We introduced RID to estimate reconstructions quality for images. It was empirically shown that this metric could perform content-based comparison of reconstructed images. Using RID, we proved the value of augmentation in our experiments. We showed that the augmented adversarial loss in this framework plays a key role in getting not only good reconstructions but good generated images.

Some open questions are still left for future work. More complex architectures may be used to achieve better IS and RID. The random shift augmentation may not the only possible choice, and other smart choices are also possible.

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

## APPENDIX A    PROOFS

### A.1    PROOF OF PROPOSITION 1 (OPTIMAL DISCRIMINATOR)

**Proposition 1.** *Given a fixed generator $G_\theta$ and a fixed encoder $E_\varphi$, the optimal discriminator $D_{\psi^*}$ is*

$$D_{\psi^*}(x, y) = \frac{r(y|x)}{r(y|x) + p_{\theta,\varphi}(y|x)} \tag{12}$$

*Proof.* For fixed generator and encoder, the value function $V(\psi)$ with respect to the discriminator is

$$V(\psi) = \mathbb{E}_{p^*(x)r(y|x)} \log D_\psi(x, y) + \mathbb{E}_{p^*(x)p_{\theta,\varphi}(y|x)} \log(1 - D_\psi(x, y)) \tag{19}$$

Let us introduce new variables and notations

$$t = (x, y), \quad p_1(t) = p^*(x)r(y|x), \quad p_2(t) = p^*(x)p_{\theta,\varphi}(y|x) \tag{20}$$

Then

$$V(\psi) = \mathbb{E}_{p_1(t)} \log D_\psi(t) + \mathbb{E}_{p_2(t)} \log(1 - D_\psi(t)) \tag{21}$$

Using the results of the paper Goodfellow et al. (2014) we obtain

$$D_{\psi^*}(t) = \frac{p_1(t)}{p_1(t) + p_2(t)} = \frac{p^*(x)r(y|x)}{p^*(x)r(y|x) + p^*(x)p_{\theta,\varphi}(y|x)} = \frac{r(y|x)}{r(y|x) + p_{\theta,\varphi}(y|x)} \tag{22}$$

$\square$

### A.2    PROOF OF PROPOSITION 2

**Proposition 2.** *The minimization of the value function $V$ under an optimal discriminator $D_{\psi^*}$ is equivalent to the minimization of the expected Jensen-Shanon divergence between $r(y|x)$ and $p_{\theta,\varphi}(y|x)$, i.e.*

$$\theta^*, \varphi^* = \arg\min_{\theta,\varphi} V(\theta, \varphi, \psi_*) = \arg\min_{\theta,\varphi} \mathbb{E}_{p^*(x)} JSD(r(y|x)\|p_{\theta,\varphi}(y|x)) \tag{13}$$

*Proof.* As in the paper Goodfellow et al. (2014) we rewrite the value function $V(\theta, \varphi)$ for the optimal discriminator $D_{\psi^*}$ as follows

$$V(\theta, \varphi) = \mathbb{E}_{p^*(x)r(y|x)} \log D_{\psi^*}(x, y) + \mathbb{E}_{p^*(x)p_{\theta,\varphi}(y|x)} \log(1 - D_{\psi^*}(x, y)) = \tag{23}$$

$$= \mathbb{E}_{p^*(x)r(y|x)} \log \frac{r(y|x)}{r(y|x) + p_{\theta,\varphi}(y|x)} + \mathbb{E}_{p^*(x)p_{\theta,\varphi}(y|x)} \log \frac{p_{\theta,\varphi}(y|x)}{r(y|x) + p_{\theta,\varphi}(y|x)} = \tag{24}$$

$$= \mathbb{E}_{p^*(x)} \left[ \mathbb{E}_{r(y|x)} \log \frac{r(y|x)}{r(y|x) + p_{\theta,\varphi}(y|x)} + \mathbb{E}_{p_{\theta,\varphi}(y|x)} \log \frac{p_{\theta,\varphi}(y|x)}{r(y|x) + p_{\theta,\varphi}(y|x)} \right] = \tag{25}$$

$$= \mathbb{E}_{p^*(x)} \left[ -\log(4) + 2 \cdot JSD\left(r(y|x)\|p_{\theta,\varphi}(y|x)\right) \right] \tag{26}$$

$\square$

## APPENDIX B    CHOICE OF AUGMENTATION

Augmentation choice might be problem specific. Thereby we additionally study different augmentations and provide an intuition how to choose the right transformation. Theory suggests to pick up a stochastic augmentation. The practical choice should take into account the desired properties of reconstructions. A random shift of an image by a small margin is sufficient to create good quality reconstructions. However, this shift should not be large because it may inherit augmentation artifacts. This can be spotted beforehand just looking at pairs $(x, a(x))$. Once these pairs are not satisfactory, model reconstructions would be bad as well.

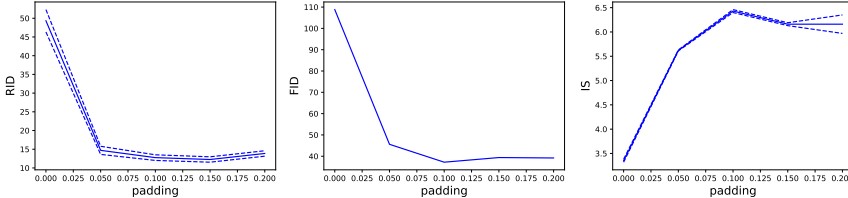

Figure 4: RID, FID, IS dependence on padding choice. Inception score suggests 0.1 padding being most robust and yielding good samples as well as reconstuctions. RID, FID grow as padding increases indicating slightly worse performance.

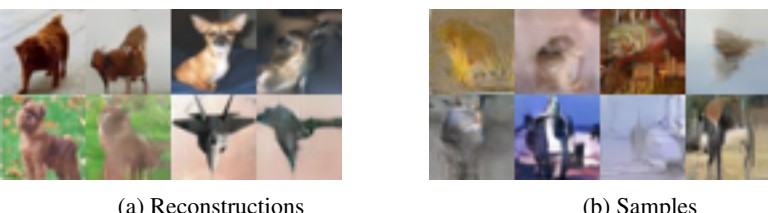

(a) Reconstructions                              (b) Samples

Figure 5: Negative effects that are caused by too aggressive augmentation (0.2) are seen in reconstructions but not in samples

## B.1 PADDING SIZE

In this experiment, we investigate the effects caused by the padding and random crop augmentation. We choose different padding size (comparatively to the original image size) and plot FID, RID and IS metrics. The results provide the intuition to choose padding size (see Figure 4). Padding should be chosen to maintain visual content while making impossible to compare augmented and original images by nearly element-wise comparison. Larger padding cause undesirable effects in reconstructions that are captured by RID (see Figure 5). Visual quality of samples, on the other hand is slightly better with more aggressive augmentation considering FID metric, what is explained by more robust training due to less mode collapse problem.

## B.2 OTHER AUGMENTATIONS

We also checked two different augmentation types: Gaussian blur and random contrast (see Figures 6,7). Both augmentations led to highly unstable training and did not yield satisfactory results (IS was 2.15 and 4.18 respectively). Therefore we conclude that a good augmentation is better to change spatial image structure preserving content (as padding does) what will force the discriminator to take content into account.

| Augmentation | | IS | FID | RID |
|---|---|---|---|---|
| crop+padding | 0 | 3.35±0.03 | 108.81 | |
| | 0.05 | 5.62±0.01 | 45.60 | 14.70±1.08 |
| | 0.1 | 6.43±0.03* (6.56±0.09) | 37.20 | 12.75±0.75 |
| | 0.15 | 6.16±0.03 | 39.38 | 12.25±0.71 |
| | 0.2 | 6.16±0.19 | 39.18 | 13.86±0.72 |
| Blur | | 2.15±0.01 | 200.66 | 32.92±1.46 |
| Contrast | | 4.18±0.01 | 101.27 | 50.02±2.10 |

Table 4: Evaluation for different types of augmentations.
*The number is taken not from the last epoch to save computational resources and compare results after the same number of epochs.

A good augmentation:

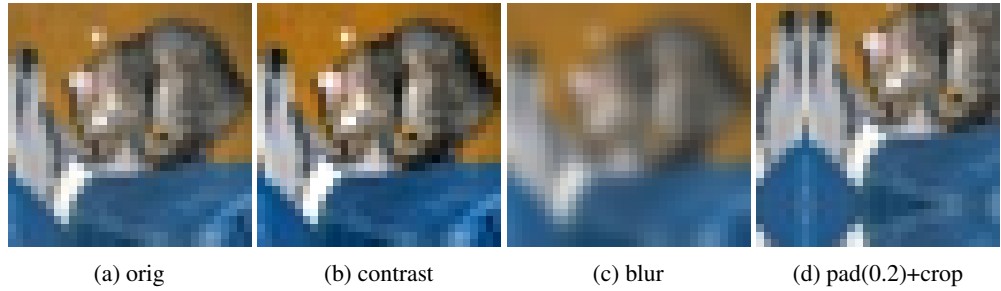

| (a) orig | (b) contrast | (c) blur | (d) pad(0.2)+crop |

Figure 6: Examples of augmentation

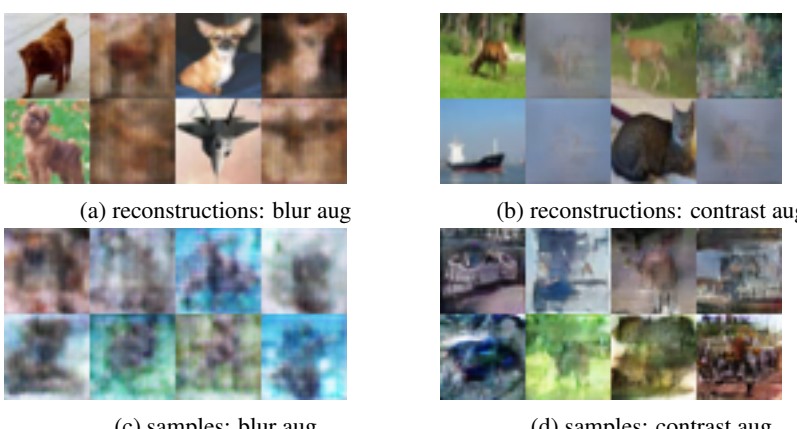

(a) reconstructions: blur aug          (b) reconstructions: contrast aug

(c) samples: blur aug          (d) samples: contrast aug

Figure 7: Instability effects that are caused by Gaussian blur and random contrast augmentations for reconstructions and samples

- is required to be non-deterministic
- should preserve the content of source image
- should be hard to use pixel-wise comparison for discriminator

## APPENDIX C    EXTENDING PAGANS

### C.1    $f$-DIVERGENCE PAGANS

$f$-GANs (Nowozin et al., 2016) are the generalization of GAN approach. Nowozin et al. (2016) introduces the model which minimizes the $f$-divergence $D_f$ (Ali & Silvey, 1966) between the true distribution $p^*(x)$ and the model distibution $p_\theta(x)$, i.e. it solves the optimization problem

$$\min_\theta D_f(p^*(x)\|p_\theta(x)) = \int p_\theta(x)f\left(\frac{p^*(x)}{p_\theta(x)}\right)dx \tag{27}$$

where $f : \mathbb{R}_+ \to \mathbb{R}$ is a convex, lower-semicontinuous function satisfying $f(1) = 0$.

The minimax game for $f$-GANs is defined as

$$\min_\theta \max_\psi V(\theta, \psi) = \mathbb{E}_{p^*(x)}T_\psi(x) - \mathbb{E}_{p_\theta(x)}f^*(T_\psi(x)) \tag{28}$$

where $V(\theta, \psi)$ is a value function and $f^*$ is a Fenchel conjugate of $f$ (Nguyen et al., 2008). For fixed parameters $\theta$, the optimal $T_{\psi^*}(x)$ is $f'\left(\frac{p^*(x)}{p_\theta(x)}\right)$. Then the value function $V(\theta, \psi^*)$ for optimal parameters $\psi^*$ equals to $f$-divergence between the distributions $p^*$ and $p_\theta$ (Nguyen et al., 2008), i.e.

$$V(\theta, \psi^*) = D_f(p^*(x)\|p_\theta(x)) \tag{29}$$

We can straightforwardly extend the definition of PAGANs model to $f$-PAGANs. We just introduce for each matching problem the $f$-GAN value function, i.e.

- generator matching:

$$\min_{\theta} \max_{\psi^{(1)}} V_f^{(1)}(\theta, \psi^{(1)}) = \mathbb{E}_{p^*(x)} T_{\psi^{(1)}}(x) - \mathbb{E}_{p_\theta(x)} f^*(T_{\psi^{(1)}}(x)) \tag{30}$$

$$\theta^* = \arg\min_{\theta} V_f^{(1)}(\theta, \psi_*^{(1)}) = \arg\min_{\theta} D_f(p^*(x)\|p_\theta(x)) \tag{31}$$

- encoder matching:

$$\min_{\varphi} \max_{\psi^{(2)}} V_f^{(2)}(\varphi, \psi^{(2)}) = \mathbb{E}_{p_z(z)} T_{\psi^{(2)}}(z) - \mathbb{E}_{q_\varphi(z)} f^*(T_{\psi^{(2)}}(z)) \tag{32}$$

$$\varphi^* = \arg\min_{\varphi} V_f^{(2)}(\varphi, \psi_*^{(2)}) = \arg\min_{\varphi} D_f(p_z(z)\|q_\varphi(z)) \tag{33}$$

- reconstruction matching:

$$\min_{\theta,\varphi} \max_{\psi} V_f(\theta, \varphi, \psi) = \mathbb{E}_{p^*(x)r(y|x)} T_\psi(x, y) - \mathbb{E}_{p^*(x)p_{\theta,\varphi}(y|x)} f^*(T_\psi(x, y)) \tag{34}$$

$$\theta^*, \varphi^* = \arg\min_{\theta,\varphi} V(\theta, \varphi, \psi_*) = \arg\min_{\theta,\varphi} D_f(r(y|x)\|p_{\theta,\varphi}(y|x)) \tag{35}$$

## C.2 WASSERSTEIN PAGANS

Arjovsky et al. (2017) proposed WGANs model for minimizing the Wasserstein-1 distance between the distributions $p^*(x)$ and $p_\theta$, i.e.

$$\min_{\theta} W(p^*(x), p_\theta(x)) = \inf_{\gamma \in \Pi(p^*, p_\theta)} \mathbb{E}_{(x,y)\sim\gamma} \|x - y\| \tag{36}$$

Because the distance $W(p^*(x), p_\theta(x))$ is intractable they consider solving the Kantorovich-Rubinstein dual problem (Villani, 2008)

$$\min_{\theta} W(p^*(x), p_\theta(x)) = \min_{\theta} \max_{\|f\|_L \leqslant 1} \left[ \mathbb{E}_{p^*(x)} f(x) - \mathbb{E}_{p_\theta} f(x) \right] \tag{37}$$

As in Section C.1 we can easily extend the PAGANs model to WPAGANs. In each matching problem the corresponding distance between distributions will be Wasserstein-1 distance.

## APPENDIX D   OTHER MODELS AND EXPERIMENT DETAILS

### D.1   TRAINING WASSERSTEIN PAGAN

As another concurrent approach to match implicit distributions we can use Wasserstein distance. Recent empirical works showed promising results (Gulrajani et al., 2017; Gemici et al., 2018) and thus they are interesting to compare with. As mentioned above we still need a critic to work on pairs of images. Unlike GAN frameworks it is desirable to have a strong critic. A channel wise concatenation for pairs $(x, y)$ worked the best in sense of visual quality and training stability. As a default choice to improve Wasserstein distance optimization we applied the gradient penalty proposed in Gulrajani et al. (2017). To apply the gradient penalty for a critic on pairs we have to interpolate between pairs $(x, y)$ and $(x', y')$. There are still two choices:

- shared alpha

$$(\tilde{x}, \tilde{y}) = (\alpha x + (1 - \alpha)x', \alpha y + (1 - \alpha)y'), \quad \alpha \sim \mathcal{U}[0, 1] \tag{38}$$

- independent alpha for each part

$$(\tilde{x}, \tilde{y}) = (\alpha_1 x + (1 - \alpha_1)x', \alpha_2 y + (1 - \alpha_2)y'), \quad \alpha_1, \alpha_2 \sim \mathcal{U}[0, 1] \tag{39}$$

Empirically we found no differences in results and in further experiments used shared alpha as a default choice. The gradient penalty strength parameter $\lambda$ was set to 10 as recommended by Gulrajani et al. (2017). We used 10 discriminator steps per 1 generator/encoder step for WPAGAN to slightly improve quality in this setting, other parameters were unchanged. In Table 5 we present results for Wasserstein loss used instead of standard GAN objective in PAGAN model. While having good reconstructions this type of loss failed to achieve good generation results.

Table 5: Inception Score and Fréchet Inception Distance for Wasserstein PAGAN.

| Model | FID | IS | RIS |
|---|---|---|---|
| WPAGAN | 52.29 | $5.62 \pm 0.09$ | $13.44 \pm 0.44$ |

## APPENDIX E    IMAGES

### E.1    PAGAN-L1 VISUAL RESULTS

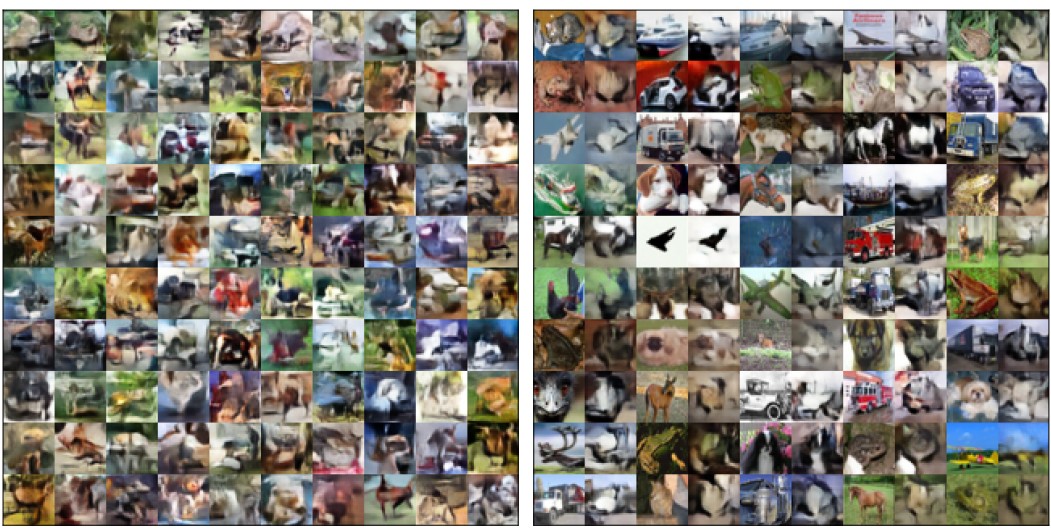

(a) CIFAR10 samples from PAGAN-L1    (b) CIFAR10 reconstructions from PAGAN-L1

Figure 8: Evaluation of Generator and Encoder trained on CIFAR10 dataset with adversarial loss replaced with $L_1$ loss. On plot (b) odd columns denote original images, even stand for corresponding reconstructions on test partition

### E.2    PAGAN-NOAUG VISUAL RESULTS

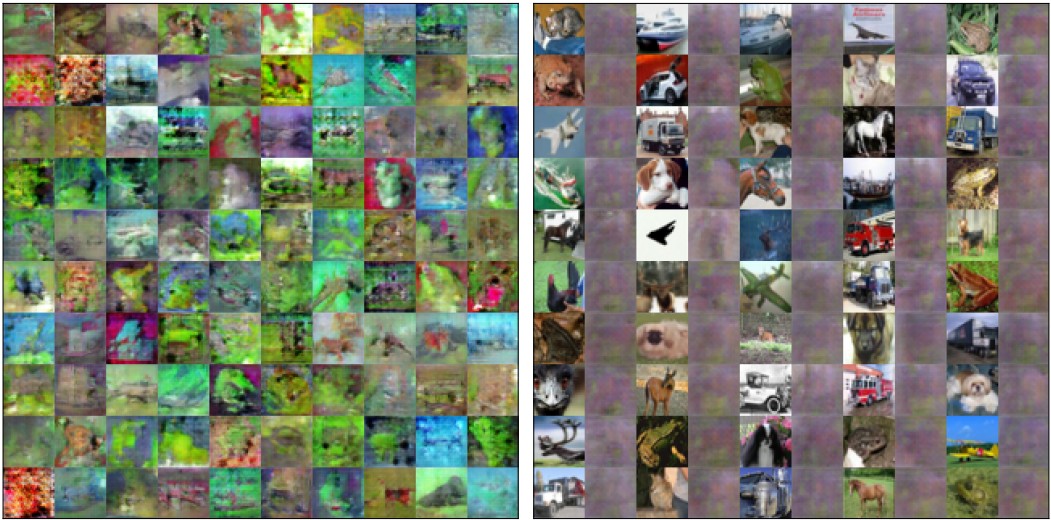

(a) CIFAR10 samples from PAGAN-NOAUG (b) CIFAR10 reconstructions from PAGAN-NOAUG

Figure 9: Evaluation of Generator and Encoder trained on CIFAR10 dataset with removed augmentation. On plot (b) odd columns denote original images, even stand for corresponding reconstructions on test partition

### E.3 PAGAN VISUAL RESULTS

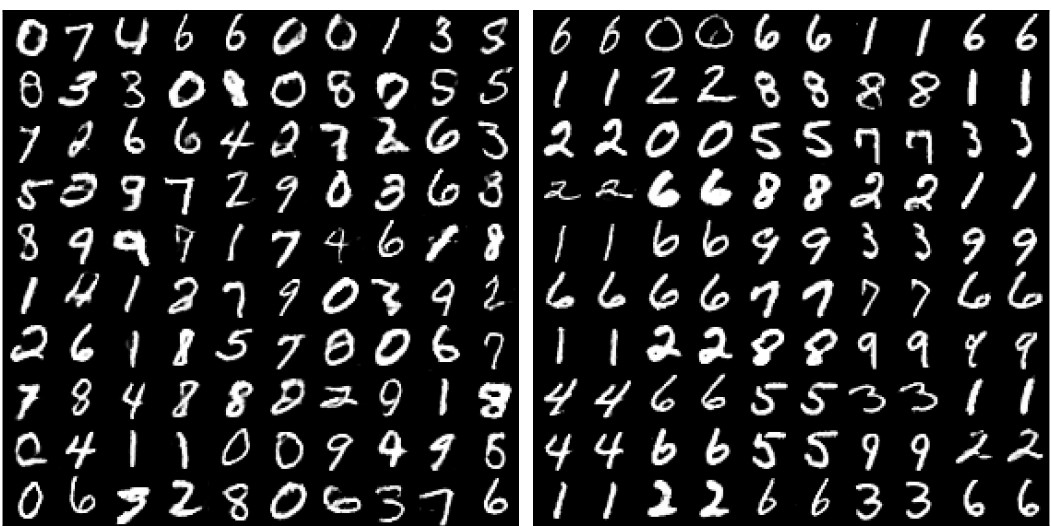

(a) MNIST samples from PAGAN (b) MNIST reconstructions from PAGAN

Figure 10: Evaluation of Generator and Encoder trained on MNIST dataset. On plot (b) odd columns denote original images, even stand for corresponding reconstructions on test partition

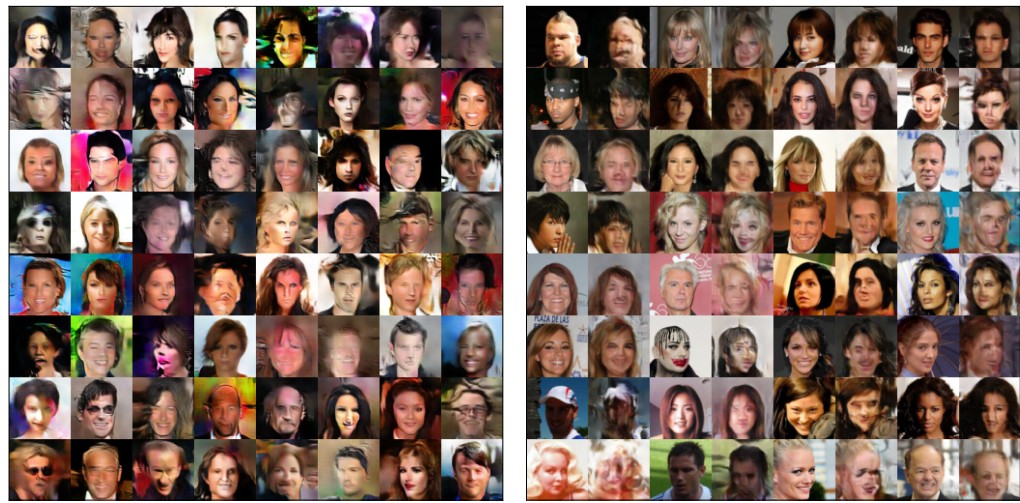

(a) celebA samples from PAGAN        (b) celebA reconstructions from PAGAN

Figure 11: Evaluation of Generator and Encoder trained on celebA dataset. On plot (b) odd columns denote original images, even stand for corresponding reconstructions on test partition

### E.4 VAE FOR RECONSTRUCTION INCEPTION SCORE

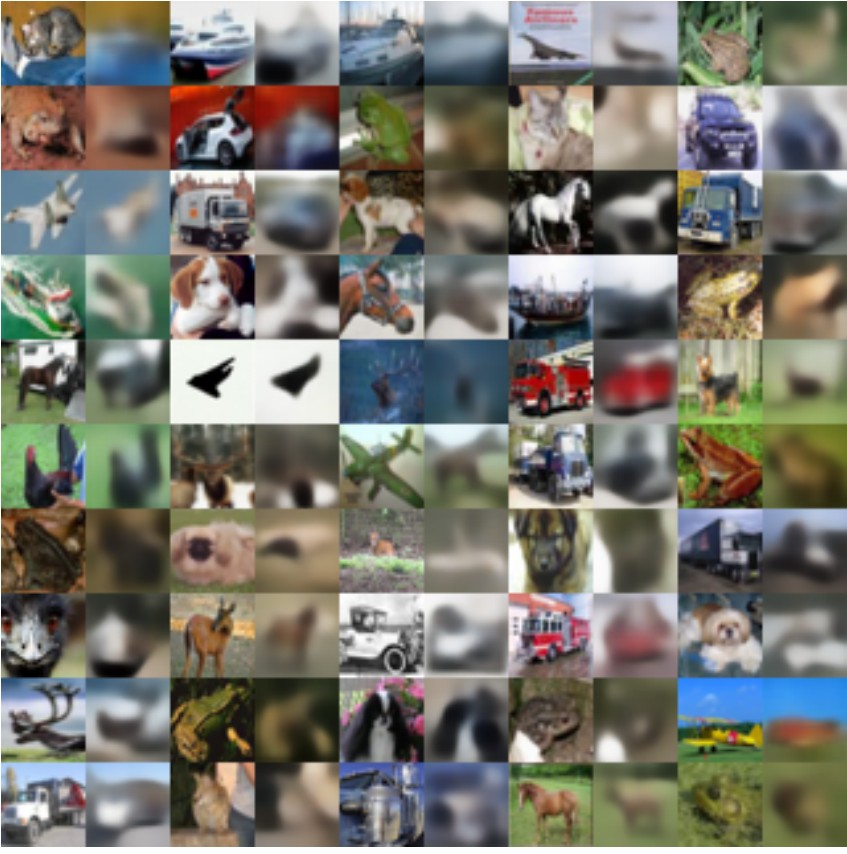

Figure 12: Reconstructions from VAE used to compute RIS

