# OpenReview forum: "PAIRWISE AUGMENTED GANS WITH ADVERSARIAL RECONSTRUCTION LOSS"
_ICLR.cc/2019/Conference_

### Official Review · AnonReviewer3 · 2018-10-26
**Quite a lot of experiments, but the choice of r(y|x) is not well justified, and some theoretical issues**

**Rating:** 5
**Confidence:** 4

**Review:**

Thank you for an interesting read.

The paper proposes adding an adversarial loss to improve the reconstruction quality of an auto-encoder. To do so, the authors define an auxiliary variable y, and then derive a GAN loss to discriminate between (x, y) and (x, decoder(encoder(x))). The algorithm is completed by combining this adversarial "reconstruction" loss with adversarial loss functions that encourages the matching of marginal distributions for both the observed variable x and the latent variable z.

Experiments present quite a lot of comparisons to existing methods as well as an ablation study on the proposed "reconstruction" loss. Improvements has been shown on reconstructing input images with significant numbers.

Overall I think the idea is new and useful, but is quite straight-forward and has some theoretical issues (see below). The propositions presented in the paper are quite standard results derived from the original GAN paper, so for that part the contribution is incremental and less interesting. The paper is overall well written, although the description of the augmented distribution r(y|x) is very rush and unclear to me.

There is one theoretical issue for the defined "reconstruction" loss (for JS and f-divergences). Because decoder(encoder(x)) is a deterministic function of x, this means p(y|x) is a delta function. With r(y|x) another delta function (even that is not delta(y=x)), with probability 1 we will have mismatched supports between p(y|x) and r(y|x).

This is also the problem of the original GAN, although in practice the original GAN with very careful tuning seem to be OK... Also it can be addressed by say instance noise or convolving the two distributions with a Gaussian, see [1][2].

I think another big issue for the paper is the lack of discussion on how to choose r(y|x), or equivalently, a(x).

1. Indeed matching p_{\theta}(x) to p^*(x) and q(z) to p(z) does not necessarily returns a good auto-encoder that makes x \approx decoder(encoder(x)). Therefore the augmented distribution r(y|x) also guides the learning of p(y|x) and with appropriately chosen r(y|x) the auto-encoder can be further improved.

2. The authors mentioned that picking r(y|x) = \delta(y = x) will result in unstable training. But there's no discussion on how to choose r(y|x), apart from a short sentence in experimental section "...we used a combination of reflecting 10% pad and the random crop to the same image size...". Why this specific choice? Since I would imagine the distribution r(y|x) has significant impact on the results of PAGAN, I would actually prefer to see an in-depth study of the choice of this distribution, either theoretically or empirically.

In summary, the proposed idea is new but straight-forward. The experimental section contains lots of results, but the ablation study by just removing the augmentation cannot fully justify the optimality of the chosen a(x). I would encourage the authors to consider the questions I raised and conduct extra study on them. I believe it will be a significant contribution to the community (e.g. in the sense of connecting GAN literature and denoising methods literature).

[1] Sonderby et al. Amortised MAP Inference for Image Super-resolution. ICLR 2017.
[2] Roth et al. Stabilizing Training of Generative Adversarial Networks through Regularization. NIPS 2017.

---

> ### Author Response · Authors · 2018-11-26
> **Response to AnonReviewer3**
>
> Dear reviewer,
> Thank you very much for the encouraging and constructive comments. We will address each your question below:
>
> > There is one theoretical issue for the defined "reconstruction" loss (for JS and f-divergences). Because decoder(encoder(x)) is a deterministic function of x, this means p(y|x) is a delta function. With r(y|x) another delta function (even that is not delta(y=x)), with probability 1 we will have mismatched supports between p(y|x) and r(y|x).
>
> It appears that there is a misunderstanding. In Section 5 we notice that encoder is stochastic and has a fully factorized Gaussian distribution. Therefore, decoder(encoder(x)) is a stochastic function, too. The conditional distribution r(y|x) is not a delta function because we use the stochastic augment mapping a(x) (the combination of reflecting pad and the random crop). As a result, supports of distributions p(y|x) and r(y|x) have non-empty intersection and Jensen-Shanon divergence is defined correctly for them.
>
> > I think another big issue for the paper is the lack of discussion on how to choose r(y|x), or equivalently, a(x).
>
> We agree with you that there should be more discussion on how to choose the augmentation function a(x) and more experiments with different type of augmentations.
>
> Indeed, the choice of the augment mapping a(x) can significantly impact PAGAN performance. We analyzed another augmentations such as a Gaussian blur and a random contrast normalization before we chose the augment function a(x) mentioned in the paper. As a result, we selected a combination of reflecting pad and the random crop based on the visual judgment of generated samples and reconstructions. We have added a new subsection "Choice of Augmentation" to Section 5 where we provide experiments with other types of augmentations as well as with different padding width for the pad-crop augmentation type. We also add some discussion on intuition behind selecting the most efficient augmentation.

---

> > ### Comment · AnonReviewer3 · 2018-11-27
> > **Reply**
> >
> > Thank you for clarification. I appreciate your efforts on including more details about the design of a(x).
> >
> > On the theoretical part, this problem cannot be easily solved.
> > 1. Even when the encoder q(z|x) is stochastic, the support of p(y|x) can still be low-dimensional if dim(z) < dim(y) and the decoder is deterministic.
> > 2. The augmentation distribution r(y|x) is not guaranteed to have full support. E.g. conditioned on x, the random-crop operator a(x) cannot generate augmentations that look like Gaussian blur.
> > 3. So now given that p(y|x) and r(y|x) do not have full support, with probability 1 they will have mismatched supports.

---

> > > ### Author Response · Authors · 2018-11-28
> > > **Thanks for the fair remark**
> > >
> > > Thank you for your clarification on this theoretical issue. Indeed, the support of p(y|x) can be low-dimensional if dim(z) < dim(y). As you noticed it can be addressed by techniques used in [1, 2].
> > >
> > > [1] Sonderby et al. Amortised MAP Inference for Image Super-resolution. ICLR 2017.
> > > [2] Roth et al. Stabilizing Training of Generative Adversarial Networks through Regularization. NIPS 2017.

---

### Official Review · AnonReviewer1 · 2018-11-01
**Adversarial reconstruction loss is an interesting idea, but the paper need more polishing**

**Rating:** 6
**Confidence:** 4

**Review:**

==============Updated=====================
The authors addressed some of my concern, and I appreciated that they added more experiments to support their argument.
Although I still have the some consideration as R3, I will raise the rating to 6.

===========================================

This paper is easy to follow. Here are some questions:

1. The argument about ALI and ALICE in the second paragraph of the introduction, “… by introducing a reconstruction loss in the form of a discriminator which classifies pairs (x, x) and (x, G(E(x)))”, however, in ALI and ALICE, they use one discriminator to classify pairs (z, x) and (z, G(z)). Therefore, “… the discriminator tends to detect the fake pair (x, G(E(x))) just by checking the identity of x and G(E(x)) which leads to vanishing gradients” is problematic. Therefore, the motivation in the introduction may be some modification.

2. The authors failed to compare their model with SVAE [1] and MINE [2], which are improved versions of ALICE. And we also have other ways to match the distribution such as Triple-GAN [3] and Triangle-GAN [4], I think the authors need to run some comparison experiments.

3. The authors should discuss more about the augment mapping a(x), i.e., how to choose a(x). I think this is quite important for this paper. At least some empirical results and analysis, for example, how inception score / FID score changes when using different choices of a(x).

4. This paper claims that the proposed method can make the training more robust, but there is no such experiment results to support the argument.

[1] chen et al. Symmetric variational autoencoder and connections to adversarial learning, AISTATS 2018.
[2] Belghazi et al, Mutual Information Neural Estimation, ICML 2018.
[3] Li et al. Triple Generative Adversarial Nets, NIPS 2017.
[4] Gan et al. Triangle generative adversarial networks, NIPS 2017.

---

> ### Author Response · Authors · 2018-11-26
> **Response to AnonReviewer1**
>
> Dear reviewer,
> We would like to thank you for your thoughtful review and valuable suggestions. We will address each your question below:
>
> 1. > Therefore, the motivation in the introduction may be some modification.
>
> It appears that there is a misunderstanding. It is true that in ALI and ALICE, they use one discriminator to classify pairs (z, x) and (z, G(z)). However, this paragraph was devoted to the ALICE paper where authors introduced an additional cycle consistency term in the eq. (8) in the form of adversarially learned discriminator on pairs (x, x) and (x, G(E(x))). Later in the Proposition 1 they showed the degeneracy of the straightforward approach: the discriminator tends to learn delta(x-G(E(x))) classification rule. One of the motivation of using the augmentation is to avoid this issue and to allow the discriminator to distinguish pairs based not on the raw pixels but on the high level features which capture the perceptual similarity of images.
>
> 2. > The authors failed to compare their model with SVAE [1] and MINE [2], which are improved versions of ALICE. And we also have other ways to match the distribution such as Triple-GAN [3] and Triangle-GAN [4], I think the authors need to run some comparison experiments.
>
> Thank you for pointing out these recent papers we missed to cite and compare to. We discuss each paper below.
>
> Triple-GAN and Triangle-GAN are semi-supervised generative adversarial models. Triple-GAN allows a class conditional generation, Triangle-GAN is applied for cross-domain joint distribution matching. However, these models do not have an encoder part and are not fully unsupervised. In our paper, we compare the proposed method only with other unsupervised bidirectional GANs. Therefore, we do not consider Triple-GAN and Triangle-GAN as our baselines.
>
> MINE is an improved version of ALICE model. Unfortunately, in the original paper, authors do not provide results for CIFAR10 dataset. It is hard to reproduce them because the source code of the method is not provided.
>
> SVAE is a generative model which improves the variational auto-encoder (VAE). It is closely related to our work. We have added quantitative comparisons with SVAE to a new version of the paper (see Table 1, Table 2).
>
> 3. >  The authors should discuss more about the augment mapping a(x), i.e., how to choose a(x). I think this is quite important for this paper. At least some empirical results and analysis, for example, how inception score / FID score changes when using different choices of a(x).
>
> Indeed, the choice of the augment mapping a(x) can significantly impact PAGAN performance. We analyzed another augmentations such as a Gaussian blur and a random contrast normalization before we chose the augment function a(x) mentioned in the paper. As a result, we selected a combination of reflecting pad and the random crop based on the visual judgment of generated samples and reconstructions. We have added a new subsection "Choice of Augmentation" to Section 5 where we provide experiments with other types of augmentations as well as with different padding width for the pad-crop augmentation type. We also add some discussion on intuition behind selecting the most efficient augmentation.
>
> 4. > This paper claims that the proposed method can make the training more robust, but there is no such experiment results to support the argument.
>
> As we understand, this question relates to the following quote from the paper: "To ensure good reconstructions, we introduce an augmented adversarial reconstruction loss ... This enforces the discriminator to take into account content invariant to the augmentation, thus making training more robust." So, we claim that the augmented pairs (x, a(x)) in contrast to pairs (x, x) allow the discriminator not to degrade to a delta function, thus making training more robust. We empirically support this argument by the experiment provided in subsection "Importance of augmentation" in Section 5 and in Table 3 where we compare the model with augmented pairs (x, a(x)) versus the model with pairs (x, x).
>
> [1] chen et al. Symmetric variational autoencoder and connections to adversarial learning, AISTATS 2018.
> [2] Belghazi et al, Mutual Information Neural Estimation, ICML 2018.
> [3] Li et al. Triple Generative Adversarial Nets, NIPS 2017.
> [4] Gan et al. Triangle generative adversarial networks, NIPS 2017.

---

### Official Review · AnonReviewer2 · 2018-11-03
**Adding and encoder for the GANs is studied. But the distinctions from existing models are not obvious.**

**Rating:** 4
**Confidence:** 3

**Review:**

The paper propose a adversary method to train a bidirectional GAN with both an encoder and decoder. Comparing to the existing works, the main contribution is the introducing of an augmented reconstruction loss by training a discriminator to distinguish the augmentation data from the reconstructed data. Experimental results are demonstrated to show the generating and reconstruction performance.

The problem studied in this paper is very important, and has drawn a lot of researchers' attentions in recent years.  However, the novelties of this paper is very limited. The techniques used to train a bidirectional GAN are very standard. The only new stuff may be is the proposed reconstruction loss defined on augmented samples and reconstructed ones. But this is also not a big contribution, seems just using a slightly different way to guarantee reconstruction.

---

> ### Author Response · Authors · 2018-11-26
> **Response to AnonReviewer2**
>
> Dear reviewer,
> We would like to thank you for the thoughtful review. The main concern you raised is about the novelty of the paper. We will address each point of the review below:
>
> > Comparing to the existing works, the main contribution is the introducing of an augmented reconstruction loss by training a discriminator to distinguish the augmentation data from the reconstructed data.
>
> The summary of our contribution is accurate at a high-level. Just to be sure that there is no misunderstanding we add the key detail: the discriminator distinguishes pairs (x, a(x)) from the pairs (x, G(E(x))) where x is a real object, a(x) is its augmentation and G(E(x)) is its reconstruction. Adding x as the first element in each pair is crucial because it ensures that reconstructions G(E(x)) will correspond to the source object x. Otherwise, if we classify just the augmentation data a(x) from the reconstructions G(E(x)) instead of pairs the auto-encoding model will not be penalized for incorrect reconstructions.
>
> > The techniques used to train a bidirectional GAN are very standard. The only new stuff may be is the proposed reconstruction loss defined on augmented samples and reconstructed ones. But this is also not a big contribution, seems just using a slightly different way to guarantee reconstruction.
>
> It is true that the key distinction of our method from other algorithms of training a bidirectional GAN is the proposed adversarial reconstruction loss defined on pairs. Despite the simplicity of the concept, to the best of our knowledge it is the first successful attempt of applying the content-aware trainable distance between two images.
>
> Other approaches [1, 2, 3, 4, 5, 6] mainly utilize standard L1 and L2 distances which lead to undesirable artifacts and blurriness in reconstructions. In ALICE [6] paper authors consider a discriminator on pairs (x, x) and (x, G(E(x)) without augmentation and mention that it degrades to a delta function which is even worse than L1 and L2. Our proposed augmentation allows for the discriminator to classify pairs based not on the pixels but on the content of the image which is invariant to the augmentation.
>
> In the paper, the main motivation of the adversarial reconstruction loss over the standard pixel-wise losses is as follows: the latter match images in the space of pixels which is highly noisy and does not capture perceptual similarity while the proposed loss matches images in the space of high level features learned by the discriminator on pairs. In experiments, we show that introducing the adversarial reconstruction loss instead of L1 distance significantly improves both the visual quality of generated images and reconstructions and standard metrics such as Inception Score and Frechet Inception Distance. Therefore, we argue that the proposed loss is conceptually very different from the standard pixel-wise losses.
>
> Additionally, we want to notice that in the paper we introduce a novel metric Reconstruction Inception Dissimilarity (RID) as alternative to the standard RMSE. We empirically show that RID is more robust to content-preserving transformations and captures perceptual similarity between source image and its reconstruction rather than a pixel-wise coincidence.
>
> [1] Variational Approaches for Auto-Encoding Generative Adversarial Networks, \\ https://arxiv.org/abs/1706.04987
> [2] It Takes (Only) Two: Adversarial Generator-Encoder Networks, https://arxiv.org/abs/1704.02304
> [3] Unpaired Image-to-Image Translation using Cycle-Consistent Adversarial Networks, https://arxiv.org/abs/1703.10593
> [4] Neural Photo Editing with Introspective Adversarial Networks, https://arxiv.org/abs/1609.07093
> [5] Wasserstein Auto-Encoders, https://arxiv.org/abs/1711.01558
> [6] ALICE: Towards Understanding Adversarial Learning for Joint Distribution Matching, https://arxiv.org/abs/1709.01215

---

### Meta-Review · Area_Chair1 · 2018-12-09
**Intersting idea that needs a bit more investigation**

**Confidence:** 3
**Recommendation:** Reject

**Metareview:**

 The paper proposes an augmented adversarial reconstruction loss for training a stochastic encoder-decoder architecture. It corresponds to a discriminator loss distinguishing between a pair of a sample from the data distribution and its augmentation and pair containing the sample and its reconstruction. The introduction of the augmentation function is an interesting idea, intensively tested in a set of experiments, but, as two of the reviewers pointed out, the paper could be improved by deeper investigation of the augmentation function and the way of choosing it, which would increase significance of the contribution.